# Safe Reinforcement Learning with Natural Language Constraints

## Abstract

In this paper, we tackle the problem of learning control policies for tasks when provided with constraints in natural language. In contrast to instruction following, language here is used not to specify goals, but rather to describe situations that an agent must avoid during its exploration of the environment. Specifying constraints in natural language also differs from the predominant paradigm in safe reinforcement learning, where safety criteria are enforced by hand-defined cost functions. While natural language allows for easy and flexible specification of safety constraints and budget limitations, its ambiguous nature presents a challenge when mapping these specifications into representations that can be used by techniques for safe reinforcement learning. To address this, we develop a model that contains two components: (1) *a constraint interpreter* to encode natural language constraints into vector representations capturing spatial and temporal information on forbidden states, and (2) *a policy network* that uses these representations to output a policy with minimal constraint violations. Our model is end-to-end differentiable and we train it using a recently proposed algorithm for constrained policy optimization. To empirically demonstrate the effectiveness of our approach, we create a new benchmark task for autonomous navigation with crowd-sourced free-form text specifying three different types of constraints. Our method outperforms several baselines by achieving 6-7 times higher returns and 76% fewer constraint violations on average. Dataset and code to reproduce our experiments are available at `https://sites.google.com/view/polco-hazard-world/`.

## 1 Introduction

Reinforcement learning (RL) has shown great promise in a variety of control problems including robot navigation (Anderson et al., 2018; Misra et al., 2018) and robotic control (Levine et al., 2016; Rajeswaran et al., 2017), where the main goal is to optimize for scalar returns. However, as RL is increasingly deployed in many real-world problems, it is imperative to ensure the safety of both agents and their surroundings, which requires accounting for constraints that may be orthogonal to maximizing returns. While there exist several safe RL algorithms (Achiam et al., 2017; Chow et al., 2019; Yang et al., 2020b) in the literature, a major limitation they share is the need to manually specify constraint costs and budget limitations. In many real-world problems, safety criteria tend to be abstract and quite challenging to define, making their specification (*e.g.,* as logical rules or mathematical constraints) an expensive task requiring domain expertise.

On the other hand, natural language provides an intuitive and easily-accessible medium for specifying constraints – not just for experts or system developers, but also for potential end users of the RL agent. For example, instead of specifying a safety constraint in the form of "`if` water `not in` `previously visited states` `then` `do not visit lava`", one can simply say "*Do not visit the lava before visiting the water.*" The key challenge lies in training the RL agent to interpret natural language and accurately adhere to the constraints during exploration and execution.

In this paper, we develop a novel framework for safe reinforcement learning that can handle natural language constraints. This setting is different from traditional instruction following, in which text instructions are used to specify goals for the agent (*e.g., "reach the key"* or *"go forward two steps"*). To effectively learn a safe policy that obeys text constraints, we propose a model consisting of two key modules. First, we use a *constraint interpreter* to encode language constraints into

intermediate vector and matrix representations–this captures spatial information of forbidden states and the long-term dependency of the past states. Second, we design a *policy network* that operates on a combination of these intermediate representations and state observations and is trained using a constrained policy optimization algorithm (*e.g.,* PCPO (Yang et al., 2020b)). This allows our agent to map the abstract safety criteria (in language) into cost representations that are amendable for safe RL. We call our approach *Policy Optimization with Language COnstraints* (POLCO).

Since there do not exist standard benchmarks for safe RL with language constraints, we construct a new navigation task called *Hazard World*. Hazard World is a 2D grid world environment with diverse, free-form text representing three types of constraints: **(1)** budgetary constraints that limit resource usage or the frequency of being in undesirable states (*e.g.,* "*The lava is really hot, so it will hurt you a lot. Please only walk on it 3 times*"), **(2)** relational constraints that specify forbidden states in relation to surrounding entities in the environment (*e.g.,* "*There should always be at least 3 squares between you and water*"), and **(3)** sequential constraints that depend on past events (*e.g.,* "*Grass will surround your boots and protect you from dangerous lava.*'). Fig. 1 provides a sample situation from the task.

In summary, we make the following key contributions. First, we formulate the problem of safe RL with safety criteria specified in natural language.

Figure 1: Learning to navigate with language constraints. The figure shows (1) three types of language constraints, (2) items which provide rewards when collected, and (3) a third-person view of the environment. The objective is to maximize total reward without violating text constraints.

Second, we propose POLCO, a new policy architecture and two-stage safe RL algorithm that first encodes natural language constraints into quantitative representations and then uses these representations to learn a constraint-satisfying policy. Third, we introduce a new safe RL dataset (Hazard World) containing three broad classes of abstract safety criteria, all described in diverse free-form text. Finally, we empirically compare POLCO against other baselines in Hazard World. We show that POLCO outperforms other baselines by achieving 6-7 times higher returns and 76% fewer constraint violations on average over three types of constraints. We also perform extensive evaluations and analyses of our model and provide insights for future improvements.

## 2    RELATED WORK

**Policy optimization with constraints.** Learning constraint-satisfying policies has been explored in prior work in the context of safe RL (see Garcia & Fernandez (2015) for a survey). Typically, the agent learns policies either by (1) exploring the environment to identify forbidden behaviors (Achiam et al., 2017; Tessler et al., 2018; Chow et al., 2019; Yang et al., 2020b; Stooke et al., 2020), or (2) through expert demonstration data to recognize the safe trajectories (Ross et al., 2011; Rajeswaran et al., 2017; Gao et al., 2018; Yang et al., 2020a). Critically, these works all require a human to specify the cost constraints manually. In contrast, we use natural language to describe the cost constraints, which allows for easier and more flexible specifications of safety constraint.

**Instruction following without constraints.** Instruction following for 2D and 3D navigation has been explored in the context of deep RL (MacMahon et al., 2006; Vogel & Jurafsky, 2010; Chen & Mooney, 2011; Artzi & Zettlemoyer, 2013; Kim & Mooney, 2013; Andreas & Klein, 2015; Thomason et al., 2020; Luketina et al., 2019; Tellex et al., 2020). Prior work either focuses on providing a dataset with a real-life visual urban or household environment (*e.g.,* Google street view) (Bisk et al., 2018; Chen et al., 2019; Anderson et al., 2018; de Vries et al., 2018); or proposes a computational model to learn multi-modal representations that fuse 2D or 3D images with goal instructions (Janner et al., 2018; Blukis et al., 2018; Fried et al., 2018; Liu et al., 2019; Jain et al., 2019; Gaddy & Klein, 2019; Hristov et al., 2019; Fu et al., 2019). These work use text to specify goals, not environmental hazards. In contrast, we use language to describe the constraints that the agent must obey.

**Constraints in natural language.** Misra et al. (2018) propose two datasets called *LANI* and *CHAI* to study spatial and temporal reasoning, as well as perception and planning. Their dataset contains a few *trajectory constraints*, which specify goal locations (*e.g.,* "*go past the house by the right*

*side of the apple*"). However, (1) their 'constraints' are actually *goal instructions* describing where the reward is–not where forbidden states are, and (2) their model first recognizes (sub)goals from language and then uses these language-defined goals as reward functions. Prakash et al. (2020) also propose a method to convert constraints in synthetic language to reward functions. These approaches do not guarantee safety since they do not explicitly model the constraints. In our work, we define 'constraints' as restrictions in state space, or specifications of forbidden states and learn safe policies.

The table below provides a quick summary of related work in instruction following and safe RL. We differentiate these methods on several key axes: (1) whether they propose a new dataset, algorithm or model, (2) whether they optimize reward functions or satisfy cost constraints, and (3) whether they have text or 3D view simulation components. We also briefly describe their main objectives.

| Work | Properties | | Objective |
|---|---|---|---|
| Touchdown (Chen et al., 2019) | 🗄 | ⭐ 🗎👁 | Dataset for multi-modal representation learning |
| LANI & CHAI (Misra et al., 2018) | 🗄 | ✿⭐ 🗎👁 | Dataset and model for multi-modal learning |
| Checker (Prakash et al., 2020) | | ✿⭐ | Convert text constraints to reward functions |
| CPO (Achiam et al., 2017) | 📄 | ⭐△ | Safe RL algorithm with a conditional gradient update |
| PCPO (Yang et al., 2020b) | 📄 | ⭐△ | Safe RL algorithm with a projection gradient update |
| **POLCO** (ours) | 🗄📄✿ | ⭐△🗎 | Framework for safe RL with language constraints |

🗄 dataset 📄 algorithm ✿ model ⭐ reward, △ constraint, 🗎 natural language 👁 3D view

## 3 PROBLEM FORMULATION

Our goal is to learn a constraint-satisfying policy that imposes safety or other application-specific constraints using text descriptions. We formulate our problem as a partially observable constrained Markov decision process (PO-CMDP) (Kaelbling et al., 1998; Altman, 1999) with text, which is defined by the tuple $< \mathcal{S}, \mathcal{O}, \mathcal{A}, T, Z, R, C, \mathcal{X} >$. Here $\mathcal{S}$ is the set of states, $\mathcal{O}$ is the set of observations, $\mathcal{A}$ is the set of actions, $T$ is the conditional probability $T(s'|s,a)$ of the next state $s'$ given the current state $s$ and action $a$, $Z$ is the conditional probability $Z(o|s)$ of the observation $o$ given the state $s$, $R : \mathcal{S} \times \mathcal{A} \to \mathbb{R}$ is the reward function, $C : \mathcal{S} \times \mathcal{A} \to \mathbb{R}$ is the true underlying constraint specification, and $\mathcal{X}$ is the set of constraint specifications in natural language. The reward function encodes the benefit of using action $a$ in state $s$, while the cost specification encodes penalties for constraint violations. The agent does not know $C$ and only observes constraints from $\mathcal{X}$. We assume that there exists a mapping from a language constraint $x \in \mathcal{X}$ to a cost specification $C$.

A policy $\pi : \mathcal{O} \to \mathcal{P}(\mathcal{A})$ is a mapping from observation $\mathcal{O}$ to the distributions of actions $\mathcal{A}$. Let $\gamma \in (0,1)$ denote a discount factor, $\mu : \mathcal{S} \to [0,1]$ denote the initial state distribution, and $\tau$ denote the trajectory $\tau = (o_0, a_0, o_1, \cdots)$ induced by a policy $\pi$. We seek a policy $\pi$ that maximizes the cumulative discounted reward $J_R$ while keeping the cumulative discounted cost $J_C$ below the cost constraint threshold $h_C(x)$:

$$\max_\pi J_R(\pi) \doteq \mathbb{E}_{\tau \sim \pi} \left[ \sum_{t=0}^\infty \gamma^t R(s_t, a_t) \right] \text{ s.t. } J_C(\pi) \doteq \mathbb{E}_{\tau \sim \pi} \left[ \sum_{t=0}^\infty \gamma^t C(s_t, a_t; x) \right] \le h_C(x), \quad (1)$$

where $\tau \sim \pi$ is shorthand for indicating that the distribution over trajectories depends on $\pi : s_0 \sim \mu, o_t \sim Z(\cdot|s_t), a_t \sim \pi(\cdot|o_t), s_{t+1} \sim T(\cdot|s_t, a_t)$, and we use $C(s_t, a_t; x)$ and $h_C(x)$ to emphasize that they are specified by $x$. And we will drop $x$ for notation simplicity.

## 4 LEARNING POLICIES WITH NATURAL LANGUAGE CONSTRAINTS

### 4.1 MODEL

We now describe our proposed neural model in POLCO, illustrated in Fig. 2. The model consists of two parts – **(1)** the *constraint interpreter* processes the text to form safety criteria (constraint mask and threshold) and **(2)** the *policy network* subsequently produces an action. For simplicity, we assume state $s$ and observation $o$ to be 2D matrices, although the model can easily be extended to other input representations (*e.g.,* we describe an extension to 3D scene inputs in the appendix).

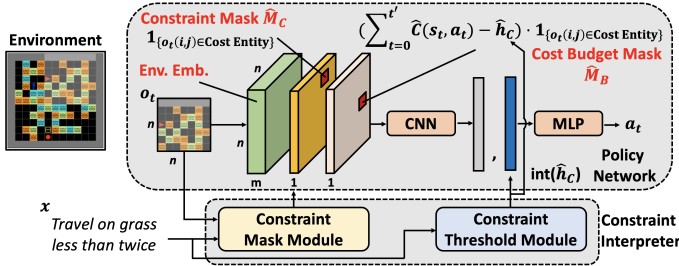

Figure 2: **Model overview.** Our model consists of two components. **(1)** The *constraint interpreter* takes a natural language constraint $x$ and an observation $o_t$ as inputs and produces a constraint mask $\hat{M}_C$ and cost constraint threshold prediction $\hat{h}_C$. **(2)** a *policy network* takes an environment embedding, a constraint mask $\hat{M}_C$, and a cost budget mask $\hat{M}_B$ that specifies cost satisfaction at each step as inputs and produces an embedding. We further concatenate this embedding with the embedding of $\hat{h}_C$, followed by an MLP to produce an action $a_t$.

**Constraint interpreter.** Fig. 3 illustrates the constraint interpreter. The constraint interpreter itself consists of two parts – **(1)** a *constraint mask module* and a **(2)** *constraint threshold module*.

**(1)** The *constraint mask* module uses the observation $o_t$ and the text $x$ to predict a binary *constraint mask*, denoted by $\hat{M}_C$, a prediction of the true constraint mask $M_C$. Each cell in $\hat{M}_C$ will contain a one if there is a cost entity (*i.e.,* the forbidden states mentioned in the text) in $i$th row and $j$th column of the observation $o_t$ (denoted by $o_t(i, j)$). Otherwise, the cell contains a zero. We use $\hat{M}_C$ to identify the cost entity in texts while preserving its spatial information for the policy network. **(2)** The *constraint threshold* module uses an LSTM to obtain the text vector representation, followed by a dense layer to produce $\hat{h}_C$, a prediction of the true constraint threshold $h_C$. Full details of the constraint interpreter can be found in Appendix A.2.

For sequential constraints with long-term dependency of the past states, $\hat{M}_C$ changes based on the states visited. For example, in Fig. 3(b), after the agent visits '*water*', $\hat{M}_C$ starts to locate the cost entity (*i.e.,* '*grass*'). Hence, we will need memory to keep track of the visited states. For this, we use an LSTM that takes the vector representation produced by a convolutional neural network (CNN) as an input and predicts $\hat{M}_C$, as illustrated in Fig. 3(b). Using $\hat{M}_C$ and $\hat{h}_C$ allows us to specify safety criteria in text while encoding them into the policy network.

**Policy network.** The policy network produces an action given the safety criteria processed by the constraint interpreter. The input to the network is the environment embedding (a tensor of size $n \times n \times m$) that encodes the observation $o_t$ which is of size $n \times n$. This tensor is concatenated with the constraint mask $\hat{M}_C$ and a *cost budget mask*, denoted by $\hat{M}_B$, a prediction of the true $M_B$. $\hat{M}_B$ is a $n \times n$ matrix, where each element takes the value of $\sum_{t=0}^{t'} \hat{C}(s_t, a_t; x) - \hat{h}_C$ (*i.e.,* the number of cost violations until $t'$th step) if there is a cost entity in $o_t(i, j)$, or zero otherwise. Note that at training and test time, we estimate the cumulative cost $\sum_{t=0}^{t'} \hat{C}(s_t, a_t; x)$ using $\hat{M}_C$ and the agent's current location at time $t$. This allows the agent to understand the constraint satisfaction at each step and use this information to accordingly plan for safe trajectories. Note that $\hat{M}_C$ is produced by the constraint interpreter, not by the policy network itself. This allows us to take advantage of the factorization of the policy network and to separately design a constraint mask module that produces an appropriate $\hat{M}_C$ for different types of constraints.[1]

After applying both constraint mask $\hat{M}_C$ and cost budget mask $\hat{M}_B$ to the environment embedding, we then feed the output into CNN to obtain a vector representation. While the constraint threshold prediction $\hat{h}_C$ (an important metric to measure safety) is an input to the constraint budget mask $\hat{M}_B$, empirically we observe its signal can be weakened by the downstream convolution operation (see Table 2(d)). To alleviate this issue, we further concatenate $\hat{h}_C$ with the vector representation outputted from the CNN. Finally, we pass the concatenated representation to a dense layer to produce

---

[1]In this work, $\hat{M}_B$ equates to a scaled up version of $\hat{M}_C$ since we assume only one constraint specification per episode, but this is not necessary in general since we may have multiple constraints over different cost entities. In that case, $\hat{M}_B$ may have different cost budgets for different cells (entities).

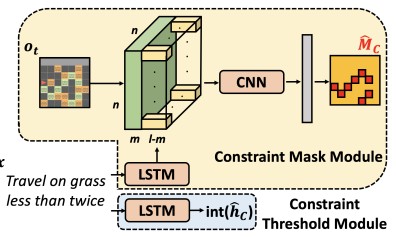
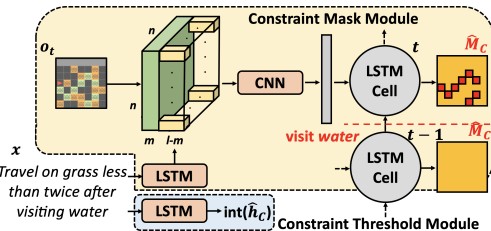

(a) For budgetary and relational constraints          (b) For sequential constraints

Figure 3: **Constraint interpreter.** **(a)** For the budgetary and relational constraints, a constraint mask module takes the environment embedding and text vector representation as inputs and predicts $\hat{M}_C$. **(b)** For the sequential constraints, we use an LSTM to store the information of the past visited states. For example, if the agent has visited '*water*', the constraint mask $\hat{M}_C$ will begin to identify the cost entity (*i.e.*, '*grass*'). In addition, for these three types of constraints, we use another LSTM given text $x$ to predict $\hat{h}_C$. (Best viewed in color.)

an action $a_t$. This policy architecture enables us to encode the language constraint into the policy network and hence learn a constraint-satisfying policy.

## 4.2 TRAINING

Training this model with RL would require significant amount of data and computation, especially when the constraint interpreter model needs to be learned from scratch. To improve data efficiency, we decompose our training procedure into two stages (Algorithm 1): **(1)** a pre-training phase for the constraint interpreter, which is followed by **(2)** a policy optimization phase to learn the rest of the policy network. Once all the model components are trained, there is no further change to the parameters. During execution, the agent is set in a random environment and provided a randomly sampled constraint to follow (*e.g., "Don't touch grass more than two times"*). We then measure the ability of the policy to complete the task without violating the constraints. Importantly, our model architecture does allow for end-to-end differentiability via tricks like Gumbel Softmax (one can treat $\hat{h}_C$ as a continuous variable). However, in this version, we do not train end-to-end for computational reasons.

---

**Algorithm 1** Learning algorithm in POLCO

**Stage 1 (Interpreter pre-training)**
    Initialize a policy $\pi^0 = \pi(\cdot|\boldsymbol{\theta}^0)$ and a buffer $\mathcal{B}$
    Run $\pi^0$ and store trajectories in $\mathcal{B}$
    Train $\Theta_1$ and $\Theta_2$ using Eq. (2) and Eq. (3)
**Stage 2 (Policy learning)**
    Empty $\mathcal{B}$
    **For** $k = 0, 1, 2, \cdots$ **do**
        Run $\pi^k = \pi(\cdot|\boldsymbol{\theta}^k)$ and store trajectories in $\mathcal{B}$
        Obtain $\boldsymbol{\theta}^{k+1}$ using Eq. (4) and Eq. (5)
        Empty $\mathcal{B}$

---

**Stage 1 (Interpreter pre-training).** For each type of constraint, we split the collected text into training and test sets using an 80-20 ratio, denoted by $\mathcal{D}_{\text{train}}$ and $\mathcal{D}_{\text{test}}$, respectively. We then form an offline pre-training dataset $\mathcal{D}_{\text{inter}}$ by using a random policy to obtain trajectories over observations $o_t$ along with corresponding text constraints $x$ from the training set $\mathcal{D}_{\text{train}}$. For the observations in $\mathcal{D}_{\text{inter}}$, we compute the corresponding ground-truth $M_C$ and $h_C$ values to use as supervision. Utilizing this dataset we first train the constraint mask module in the constraint interpreter by minimizing the following binary cross-entropy loss:

$$\mathcal{L}(\Theta_1) = -\mathbb{E}_{(o_t,x)\sim\mathcal{D}_{\text{inter}}}\left[\frac{1}{|M_C|}\sum_{i,j=1}^{n}M_C(i,j;o_t,x)\log\hat{M}_C(i,j;o_t,x)\right.$$

$$\left. + (1 - M_C(i,j;o_t,x))\log(1 - \hat{M}_C(i,j;o_t,x))\right], \quad (2)$$

where $\Theta_1$ is the parameter of the constraint mask module, $M_C(i,j;o_t,x)$ denotes the ground-truth (binary) mask label in $i$th row and $j$th column of the $n \times n$ environment given $o_t$ and $x$, and $\hat{M}_C(i,j;o_t,x)$ is the corresponding binary probability prediction of constraint mask model (here we emphasize that $M_C$ is specified by $o_t$ and $x$).

For the constraint threshold module, we minimize the following mean-square-error loss using the data from $\mathcal{D}_{\text{train}}$:

$$\mathcal{L}(\Theta_2) = \mathbb{E}_{(o_t,x)\sim\mathcal{D}_{\text{inter}}}\left[(h_C(x) - \hat{h}_C(x))^2\right], \quad (3)$$

where $\Theta_2$ is the parameter of the constraint threshold module. Unlike RL, empirically we only need a smaller dataset to train both models in the constraint interpreter, as the models tend to converge relatively quickly.

**Stage 2 (Policy learning).** We then use a state-of-the-art safe RL algorithm–projection-based constrained policy optimization (PCPO) (Yang et al., 2020b)–to train the policy network. During training, the agent interacts with the environment to obtain rewards and penalty costs from trained $\hat{M}_C$ for computing $J_R(\pi)$ and $J_C(\pi)$. PCPO is an iterative method that performs two key steps in each iteration:

**PCPO Step 1 (Optimize the reward).** The PCPO algorithm performs one step of trust region policy optimization (TRPO (Schulman et al., 2015a)) to maximize the reward advantage function $A_R^\pi(s, a)$ over a KL-divergence neighborhood of $\pi^k$:

$$\pi^{k+\frac{1}{2}} = \arg\max_\pi \ \mathbb{E}_{s \sim d^{\pi^k}, \ a \sim \pi}[A_R^{\pi^k}(s, a)] \quad \text{s.t. } \mathbb{E}_{s \sim d^{\pi^k}}\left[D_{\text{KL}}(\pi(s)\|\pi^k(s))\right] \leq \delta, \tag{4}$$

where $d^{\pi^k}$ is the state visitation frequency induced by the policy $\pi$ at $k$th update.

**PCPO Step 2 (Project to satisfy the cost constraint).** Next, PCPO projects $\pi^{k+\frac{1}{2}}$ onto the set of policies satisfying the cost constraint to minimize the distance function $D$ (*e.g.,* KL divergence):

$$\pi^{k+1} = \arg\min_\pi \ D(\pi, \pi^{k+\frac{1}{2}}) \quad \text{s.t. } J_C(\pi^k) + \frac{1}{1-\gamma}\mathbb{E}_{s \sim d^{\pi^k}, \ a \sim \pi}[A_C^{\pi^k}(s, a)] \leq \hat{h}_C, \tag{5}$$

where $\gamma$ is the discounted factor and $A_C^\pi(s, a)$ is the cost advantage function. Finally, during test time, we evaluate our model in $\mathcal{D}_{\text{test}}$.

While our method does require pre-training of the constraint interpreter with ground-truth constraint maps, we envision in future work the dependence on this phase can be reduced by leveraging techniques such as curriculum learning or pre-trained language models like BERT (Devlin et al., 2019).

## 5 EXPERIMENTS

**Task.** We construct a new task Hazard World for our experimental evaluation (Fig. 1). In this task, for each episode, the agent starts at a random location within a procedurally generated environment and receives an abstract constraint specified by natural language, sampled from a pool of all available constraints. The agent's objective is to collect all the reward entities and navigate safely (*i.e.,* avoid cost entities) by adhering to the constraint. The agent needs to satisfy the constraints during both training and testing, unlike prior work (Misra et al., 2018) that allows the agent to explore the entire state space during training without any constraints.

We implement Hazard World on top of the 2D GridWorld layout of BabyAI (Chevalier-Boisvert et al., 2018a;b). We randomly place three *reward entities* on the map: '*ball,*' '*box,*' and '*key,*' with rewards of 1, 2, and 3, respectively. We also randomly place numerous *cost entities* on the map: '*lava,*' '*water,*' and '*grass*'. A textual constraint is then imposed upon one of these cost entities (*e.g., "Touch water fewer than once"*). Note that to construct the Hazard World task we can generate multiple maps given the same text constraint. The environment $s_t$ is of size 13×13, including the surrounding walls, and the agent's observation $o_t$ is of size 7×7.

Hazard World contains three types of constraints – (1) *Budgetary constraints*, which limit on the number of times a set of states can be visited (*i.e.,* up to $h_C$), (2) *Relational constraints*, which define a minimal distance that must be maintained between the agent and a set of cost entities, and (3) *Sequential constraints*, which restrict the set of safe states, given that some condition has been met. For relational and sequential constraints, we set budget $h_C$ as zero for simplicity.

We collect free-form text in English for the constraints using Amazon Mechanical Turk (AMT). To generate a sample constraint, we provided workers with a description of Hazard World, the cost entity to be avoided, and one of three possible additional pieces of information, depending on the constraint type: the cost budget (budgetary), the minimum safe distance (relational), or the other cost entity impacted by past events (sequential). Workers were then

|       | Count | Vocab. Size | Mean Length |
|-------|-------|-------------|-------------|
| Bud.  | 432   | 274         | $9.09 \pm 4.30$  |
| Rel.  | 262   | 180         | $9.02 \pm 3.65$  |
| Seq.  | 290   | 241         | $10.36 \pm 3.49$ |
| Total | 984   | 526         | $9.44 \pm 3.95$  |

Table 1: Statistics in Hazard World.

asked to instruct another person to safely navigate Hazard World. After collecting all responses from AMT, we filtered the data to remove off-topic responses. For examples of text and AMT worker prompts, we refer the reader to Appendix A.1. Table 1 provides statistics on the text.

|  | $h_C = 0$ | | | $h_C = 2$ | | | $h_C = 4$ | | |
|---|---|---|---|---|---|---|---|---|---|
| Method | $J_R(\pi)\uparrow$ | $\Delta_C\downarrow$ | $\Delta_{RC}\uparrow$ | $J_R(\pi)\uparrow$ | $\Delta_C\downarrow$ | $\Delta_{RC}\uparrow$ | $J_R(\pi)\uparrow$ | $\Delta_C\downarrow$ | $\Delta_{RC}\uparrow$ |
| RW | 1.2 | 17.8 | -16.6 | 1.2 | 15.8 | -14.6 | 1.2 | 13.8 | -12.6 |
| CF w/ TRPO | 3.6 | 12.1 | -8.5 | 3.5 | 10.2 | -6.7 | 3.6 | 8.7 | -5.1 |
| CF w/ PCPO | 0.5 | 4.0 | -3.5 | 0.5 | 2.3 | -1.8 | 0.5 | 0.2 | 0.3 |
| PN w/ TRPO | 5.7 | 8.0 | -2.3 | 5.7 | 7.1 | -1.3 | 5.7 | 2.4 | 3.3 |
| PN w/ FPO | 0.3 | 0.1 | 0.2 | 0.2 | -0.8 | 0.2 | 0.3 | -3.8 | 0.3 |
| POLCO (ours) | 5.0 | 1.9 | **3.1** | 5.0 | 0.3 | **4.7** | 5.1 | -1.1 | **5.1** |

(a) Budgetary Constraints

|  | $h_C = 0$ | | |
|---|---|---|---|
| Method | $J_R(\pi)\uparrow$ | $\Delta_C\downarrow$ | $\Delta_{RC}\uparrow$ |
| RW | 0.9 | 19.1 | -18.2 |
| CF w/ TRPO | 2.2 | 11.2 | -9 |
| CF w/ PCPO | 0.0 | 0.0 | 0 |
| PN w/ TRPO | 5.1 | 17.3 | -12.2 |
| PN w/ FPO | 0.4 | 0.0 | 0.4 |
| POLCO (ours) | 1.1 | 0.0 | **1.1** |

(b) Relational Constraints

|  | $h_C = 0$ | | |
|---|---|---|---|
| Method | $J_R(\pi)\uparrow$ | $\Delta_C\downarrow$ | $\Delta_{RC}\uparrow$ |
| RW | 1.1 | 9.8 | -8.7 |
| CF w/ TRPO | 0.0 | 0.0 | 0.0 |
| CF w/ PCPO | 3.4 | 11.3 | -7.9 |
| PN w/ TRPO | 5.8 | 2.5 | 3.3 |
| PN w/ FPO | 0.0 | 4.5 | -4.5 |
| POLCO (ours) | 5.1 | 1.0 | **4.1** |

(c) Sequential Constraints

|  | $h_C = 0$ | | $h_C = 2$ | | $h_C = 4$ | |
|---|---|---|---|---|---|---|
| Method | $J_R(\pi)\uparrow$ | $\Delta_C\downarrow$ | $J_R(\pi)\uparrow$ | $\Delta_C\downarrow$ | $J_R(\pi)\uparrow$ | $\Delta_C\downarrow$ |
| CF w/ PCPO | 0.5 | 4.0 | 0.5 | 2.3 | 0.5 | 0.2 |
| w/o $M_C$ | 3.9 | 2.5 | 4.1 | 0.8 | 4.0 | -1.8 |
| w/o $h_C$ emb. | 4.5 | 1.6 | 4.6 | **0.2** | 4.6 | -1.2 |
| w/o $M_B$ | 5.2 | 2.3 | 5.2 | **0.2** | 5.2 | -1.6 |
| Full Model | **5.4** | **1.5** | **5.5** | **0.2** | **5.5** | **-0.9** |

(d) Ablation studies on budgetary constraints

Table 2: **(a-c)** Performance of POLCO (*i.e.,* PN w/ PCPO) and baselines across three types of constraints on the test set. **(d)** Ablations showing the effect of each component in POLCO. POLCO outperforms all the other variants by achieving higher reward and lower constraint violations. (Arrows denote higher or lower scores.)

**Train-test split.** For each type of constraint, we split the collected text into training and test sets using an 80-20 ratio, denoted by $\mathcal{D}_{\text{train}}$ and $\mathcal{D}_{\text{test}}$, respectively.

**Stage 1 (Interpreter pre-training).** To form $\mathcal{D}_{\text{inter}}$ for pre-training, we first sample a text constraint $x$ from $\mathcal{D}_{\text{train}}$ and generate a map. We then use a random policy to obtain the trajectories with rollout length 200 over observations $o_t$ along with corresponding text constraints $x$. In total, we repeat this procedure 5,000 times to form $\mathcal{D}_{\text{inter}}$ (*i.e.,* each constraint is paired with 6.35 maps on average). We train one constraint interpreter for each type of constraint, which ensures that the constraint interpreter can robustly produce $M_C$ and $h_C$. During testing of the cost constraint interpreter, we use the same procedure and generate 5,000 constraint-map pairs.

**Stage 2 (Policy learning).** For training POLCO, we use text constraints from $\mathcal{D}_{\text{train}}$ paired with randomly generated maps. Note that for each type of constraints, we train a single network with different values of $h_C$. During testing, we randomly sample constraints from $\mathcal{D}_{\text{test}}$, followed by randomly generating the map. In total, we obtain 5,000 constraint-map pairs to test the model (*i.e.,* on average one constraint is paired with 25.38 maps). Please read Appendix A.2 for more details.

**Baselines.** The baselines are combinations of the following models and training algorithms:

**(1) Models.** We consider a baseline model from Walsman et al. (2018), which does not have the $M_C$, $M_B$ and $h_C$ representations. This model simply takes a concatenation of the observations and text representations as input and produces an action. We call this model *constraint-fusion* (CF).

**(2) Algorithms. (a)** *TRPO* (Schulman et al., 2015a). In TRPO, the agent *ignores* all constraints and only optimizes the reward. **(b)** *Fixed-point policy optimization (FPO)* (Achiam et al., 2017). In FPO, the reward objective is combined with the weighted cost objective (*i.e.,* receive the penalty of 1 if the agent visits the cost entity). We include it to demonstrate that simply treating the cost penalty as a negative reward will hinder the exploration of the agent.

Finally, we also include a random walk (RW) baseline to serve as a *lower* bound on the reward and cost performance. POLCO is a combination of our policy network (PN) and the PCPO algorithm.

**Evaluation metrics.** Our primary evaluation metrics are **(1)** the average value of the reward (*i.e.,* $J_R(\pi)$), **(2)** the average cost violations $\Delta_C := J_C(\pi) - h_C$, and **(3)** the difference between the reward and the cost violations $\Delta_{RC} := J_R(\pi) - \alpha \cdot \max(0, \Delta_C)$, where $\alpha$ represents a tradeoff between reward and cost violation. We set $\alpha$ to 1 for simplicity, but $\alpha$ may depend on the task. Note that we want to enforce $\Delta_C$ to be non-positive while maximizing $J_R(\pi)$ (*i.e.,* maximize $\Delta_{RC}$).

**Overall performance.** Table 2(a-c) details the performance of POLCO and the baselines over three types of constraints on the test set. We observe from Table 2(a) that POLCO achieves the best $\Delta_{RC}$ over different choices of $h_C$, outperforming other baseline methods. For instance, when $h_C = 0$ the constraint violation of the policy network trained with TRPO is 3.21 times larger than that of POLCO. This implies that simply optimizing the reward as in Misra et al. (2018) cannot guarantee safety. We also observe that PN trained with TRPO achieves better reward performance compared when with CF and TRPO, which indicates that our proposed policy network is easier to optimize than the CF model. CF with PCPO also fails to improve the reward since merely combining the observations and the text is not sufficient to learn an effective representation for parsing the

| | Budgetary | | Relational | | Sequential | |
|---|---|---|---|---|---|---|
| | Ours | Rule | Ours | Rule | Ours | Rule |
| $M_C$ ACC | **96.06**% | 95.86% | **81.46** | 65.49% | **91.84**% | 91.36% |
| $M_C$ AUC | 98.17% | - | 90.71 | - | 96.99% | - |
| $h_C$ MSE | **0.21** | 0.74 | **0.04** | 0.52 | **0.0002** | 0.22 |

(a) Performance of the interpreter

After walking on grass do not walk on lava

Before visiting grass — Agent visits grass — After visiting grass

0.0 0.2 0.4 0.6 0.8 1.0 Probability

(b) Visualization of the constraint mask

Figure 4: **(a)** The cell-level accuracy (ACC) and area-under-curve (AUC) of predicting $M_C$, and mean-square-error (MSE) of predicting $h_C$ over three types of constraints on the test set. We observe that POLCO achieves superior results compared to the rule-based baseline. **(b)** Visualization of the constraint mask $M_C$ for the sequential constraints. We show the probability of predicting the cost entity. Our model can accurately locate the cost entity given the language constraints. The first two views are mostly black because the agent spawned facing a wall. (Best viewed in color.)

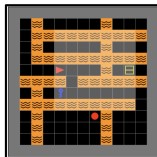

(a) Lavawall (Chevalier-Boisvert et al., 2018a)

| | $h_C = 0$ | | | $h_C = 2$ | | | $h_C = 4$ | | |
|---|---|---|---|---|---|---|---|---|---|
| | $J_R(\pi)\uparrow$ | $\Delta_C\downarrow$ | $\Delta_{RC}\uparrow$ | $J_R(\pi)\uparrow$ | $\Delta_C\downarrow$ | $\Delta_{RC}\uparrow$ | $J_R(\pi)\uparrow$ | $\Delta_C\downarrow$ | $\Delta_{RC}\uparrow$ |
| RW | 0.1 | 23.4 | -23.3 | 0.1 | 21.4 | -21.3 | 0.1 | 19.4 | -19.3 |
| CF w/ TRPO | 3.4 | 35.6 | -32.2 | 3.4 | 33.6 | -30.2 | 3.4 | 31.6 | -28.2 |
| CF w/ PCPO | 0.7 | 7.4 | -6.7 | 0.7 | 5.4 | -4.7 | 0.7 | 3.4 | -2.7 |
| PN w/ TRPO | 4.5 | 54.7 | -50.2 | 4.5 | 52.7 | -48.2 | 4.5 | 50.7 | -46.2 |
| PN w/ FPO | 1.8 | 0.0 | **1.8** | 1.8 | -2.0 | **1.8** | 1.8 | -4.0 | 1.8 |
| Ours | 3.7 | 2.9 | 0.8 | 3.9 | 2.1 | **1.8** | 4.5 | 1.8 | **2.7** |

(b) Generalization results in Lavawall

Figure 5: **(a)** Lavawall environment: Reward function and transition dynamics are same as Hazard World, but map contains only 'lava' entities. **(b)** Generalization performance in Lavawall over the tested models and algorithms. (Arrows denote higher or lower scores being better.)

constraints. Finally, the PN agent trained with FPO has a low reward, because simply treating the cost penalty as the negative reward hinders the agent's exploration.

Tables 2(b-c) also show that POLCO achieves the best $\Delta_{RC}$ for the more complex relational and sequential constraints. For the relational case, although the CF agent trained with PCPO satisfies the constraints, it has a relatively low reward. For the sequential constraints, even though the PN agent trained with TRPO obtains a higher reward than POLCO, it also has 1.5 times higher cost violations.

**Ablation studies.** We perform ablation studies on the proposed POLCO policy network to examine the importance of each component – $M_C$, $M_B$ and $h_C$ embedding. The performance over different $h_C$ in terms of the reward and cost violation is shown in Table. 2(d). To eliminate the prediction error of the constraint interpreter, here we use the *ground-truth* $M_C$ and $h_C$ instead. We observe that the full model achieves the best performance in all cases, averaging 5.12% more reward and 2.22% fewer cost violations. Without $M_C$, the agent cannot recognize the cost entities. This causes the agent to incur 66.67% higher $\Delta_C$ compared with the full model. In addition, $\Delta_C$ in the full model is almost zero. This suggests that with the $h_C$ embedding and the $M_B$ mask the agent is able to better understand the cost satisfaction at every step and hence plan for safer trajectories. In contrast, without these components the agent fails to understand the text constraints and improve the reward.

**Constraint interpreter.** Next, we examine the performance of the constraint interpreter. The result is illustrated in Fig. 4(a). Please read the caption for more details. The rule-based baseline here is similar to a $n$-gram model, in which we compute the likelihood of the cost entity given the text. Please read Appendix A.2 for more details. We observe that our model achieves superior performance in all cases. We further visualize the predicted $M_C$ for the sequential constraints shown in Fig. 4(b). We observe that after visiting 'grass', $M_C$ changes from near zero prediction probability to predict cost entities precisely. This shows that our constraint interpreter model can effectively maintain the long-term dependency of the past states and produce an accurate $M_C$.

**Generalization to different environments.** Finally, we examine whether POLCO is robust to covariate shift in the environment distribution (*i.e.,* a new environment containing only a subset of all cost entities). We first train POLCO and the other baselines in Hazard World and then test them on the commonly-used *Lavawall* (Chevalier-Boisvert et al., 2018a) task as shown in Fig. 5(a). We select free-form constraints from Hazard World imposed over the 'lava' cost entity to be the training set. Fig. 5(b) shows the reward and cost violation over different $h_C$ of different agents in the budgetary-constrained Lavawall experiment. We observe that in experiments with a larger $h_C$, generally the agents are less constrained during navigation and yield higher reward and lower cost violations.

Still, in most cases the baseline agents have substantially higher cost violations or obtain minimal reward. This shows that the POLCO agents generalize better to unseen environments, obtaining higher reward and lower constraint violations.

## 6    CONCLUSION

We addressed the problem of safe reinforcement learning when safety constraints are specified by natural language. We proposed the two-stage POLCO algorithm that both learns to understand the abstract language constraints and effectively solves for a safe policy. To demonstrate the effectiveness of POLCO, we created a new benchmark navigation task via crowd-sourcing which contains a variety of text constraints. Experiments showed that POLCO achieves superior reward performance, lower constraint violation, and better generalizability than other safe RL approaches. Future directions include **(1)** applying POLCO to 3D navigation tasks with high-dimensional (image) observations, **(2)** improving representation learning to handle more complex language constraints, and **(3)** applying POLCO to large-scale safety-critical problems (*e.g.,* robotics).

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

## A    APPENDIX

**Outline.** The appendix is outlined as follows. Section A.1 details the dataset and the procedure of collecting the dataset. Section A.2 describes the parameters of the constraint interpreter and the policy network, and the PCPO training details. Section A.3 provides the learning curves of training the policy network. Finally, we fill the Machine Learning Reproducibility Checklist in Section A.5.

| Constraint Type | Examples |
|---|---|
| Budgetary | The water should only be stepped on a max of 5 times. |
| | Lava hurts a lot, but you have special shoes that you can use to walk on it, but only up to 5 times, remember! |
| | You can get in lava, but only once. |
| | Four is the most number of times you can touch water |
| | You cannot step on the lava block at all. You will die otherwise. |
| Relational | Water will hurt you if you are two steps or less from them. |
| | Always stay 1 step away from lava |
| | Any block within one unit of a grass cannot be touched. |
| | The explosion radius of grass is three, so stay at least that distance away from grass. |
| | Waters are dangerous, so do not cross them. |
| Sequential | Make sure you don't walk on water after walking on grass. |
| | Do not touch the water or water will become risky. |
| | You may touch the water first, but the lava is dangerous so do not touch it after. |
| | Avoid lava since you can only walk on it once. After that the lava will hurt you. |
| | Water will trigger grass to become dangerous. |
| Invalid | good |
| | move foreward |
| | Just avoid the perimeter when collecting the objects, and you'll be safe. |
| | Your directions are as follows: if you're facing a block with a water block in front of it, walk five blocks ahead … (81 more words) |
| | asdf |

Table 3: Examples from the various constraint classes. When a constraint does not fully describe all forbidden states in the environment, we classify it as invalid.

## A.1 DATASET

At a high level, Hazard World applies the instruction following paradigm to safe reinforcement learning. Concretely, this means that safety constraints in our environment are specified via language. Our dataset is thus comprised of two components: the environment, made up of the objects that the agent interacts with, and the constraint, which imposes a restriction on which environmental states can be visited.

The environment is procedurally generated. For each episode, Hazard World places the agent at a randomized start location and fills the environment with objects. Hazard World then randomly samples one constraint out of all possible constraints and assigns this constraint to the environment.

We collected natural language constraints in a two-step process. In the first step, or the data generation step, we prompted workers on Amazon Mechanical Turk with scenarios shown in Fig. 6. Workers are provided the minimum necessary information to define the constraint and asked to describe the situation to another person. For example, to generate a so-called budgetary constraint, workers are given the cost entity to avoid (*'lava', 'grass'* or *'water'*) and the budget (*i.e.,* $h_C$, a number 0 through 5). The workers use this information to write an instruction for another person. This allows us to ensure that the texts we collected are free-form. These generations form our language constraints.

In the second step, or the data validation step, we employed an undergraduate student to remove invalid constraints. We define a constraint as invalid if (a) the constraint is off-topic or (b) the constraint does not clearly describe states that should be avoided. Examples of valid and invalid constraints are included in Table 3. Finally, we randomly split the dataset into 80% training and 20% test sets.

In Hazard World and Lawawall, the agent has 4 actions in total: $a \in \mathcal{A} = \{\text{right}, \text{left}, \text{up}, \text{down}\}$. The transition dynamics $T$ is deterministic.

Here, your friend is navigating the environment below to collect the ball, box, and key. The environment also contains lava, water, and grass blocks.

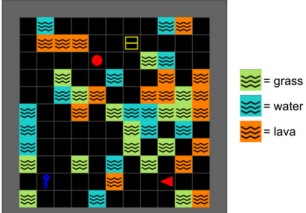

= grass
= water
= lava

(a) General prompt for all constraint classes.

One of these block types causes wear and tear, so your friend can only stand on them a limited number of times. For example, if **Number of Times = 3**, then your friend can only walk on the **Dangerous Block** three times, at most. Knowing this, tell the other person what she can or cannot do in this situation.

**Dangerous Block:** lava

**Number of Times:** 5

Type what you would say here...

(b) Budgetary prompt.

One of these objects is dangerous, and your friend must maintain a minimum safe distance from that type of object. For example, if **Minimum Distance = 1**, then your friend must stay at least 1 step away from **Dangerous Blocks** at all times. Knowing this, tell the other person what she can or cannot do in this situation.

**Dangerous Block:** lava

**Minimum Distance:** 3

Type what you would say here...

(c) Relational prompt.

One of the three block types is a trigger block. If you friend touches a trigger block, one of the block types becomes dangerous. Your job is to tell the other person what she can or cannot do in this situation. For example, if **Trigger Block = lava**, then your friend can't walk on **Dangerous Blocks** after walking on any **lava** blocks.

**Trigger Block:** lava

**Dangerous Block:** water

Type what you would say here...

(d) Sequential prompt.

Figure 6: AMT workers receive the general prompt and one of the three specific prompts. They are then asked to instruct another person for the given situation. This ensures that the texts we collected are free-form.

## A.2 ARCHITECTURES, PARAMETERS, AND TRAINING DETAILS

**Policy network in POLCO.** The architecture of the policy network is shown in Fig. 7. The environment embedding for the observation $o_t$ is of the size $7 \times 7 \times 3$. This embedding is further concatenated with the cost constraint mask $M_C$ and the cost budget mask $M_B$. This forms the input with the size $7 \times 7 \times 5$. We then use convolutions, followed by dense layers to get a vector with the size 5. This vector is further concatenated with the $h_C$ embedding. Finally, we use dense layers to the categorical distribution with four classes (*i.e.,* turn right, left, up or down in Hazard World). We then sample an action from this distribution.

**Constraint interpreter in POLCO.** The architecture of the constraint interpreter is shown in Fig. 8. For the constraint mask module, the input is the text with $w$ words. We then use an embedding network, followed by an LSTM to obtain the text embedding with the size 5. The text embedding is duplicated to get a tensor with the size $7 \times 7 \times 5$. This tensor is concatenated with the observation of size $7 \times 7 \times 3$, creating a tensor with the size $7 \times 7 \times 8$. In addition, we use a convolution, followed by dense layers and a reshaping to get the cost constraint mask $M_C$.

Next, we use a heuristic to compute $\hat{C}_{tot} := \sum_{t=0}^{t'} C(s_t, a_t; x)$ from $M_C$. At execution time, we give our constraint interpreter access to the agent's actions. We initialize $\hat{C}_{tot} = 0$. Per timestep,

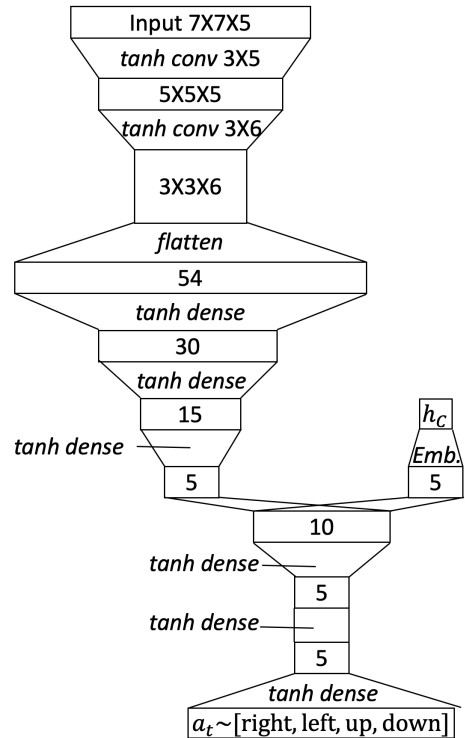

Figure 7: Description of the policy network in POLCO.

| Parameter | |
|---|---|
| Reward dis. factor $\gamma$ | 0.99 |
| Constraint cost dis. factor $\gamma_C$ | 1.0 |
| step size $\delta$ | $10^{-3}$ |
| $\lambda_R^{\text{GAE}}$ | 0.95 |
| $\lambda_C^{\text{GAE}}$ | 0.9 |
| Batch size | 10,000 |
| Rollout length | 200 |
| Number of policy updates | 2,500 |

Table 4: Parameters used in POLCO.

our agent either turns or moves forward. If the agent moves forward and the square in front of the agent contains a cost entity according to $M_C$, we increment $\hat{C}_{tot}$.

For the constraint threshold module, we use the same architecture to get the text embedding. We then use dense layers to predict the value of $h_C$.

**Rule-based baseline.** The rule-based baseline is similar to a $n$-gram model, in which we compute the likelihood of the cost entity given the text. Specifically, in the training set we count the number of *word-cost entity* pairs. For example, for each map, given the language constraints "*The water should only be stepped on a max of 5 times*" and the ground-truth cost entity (*e.g., 'water'*), the pair '*the*'-'*water*' is added one and so forth. During testing, for a given text and map, we compute the following probability for each possible cost entity:

$$P(\text{'water'}|\text{text constraints}) \propto P(\text{text constraints})P(\text{'water'})$$

$$=\Pi_{\text{word}}P(\text{word}|\text{'water'})P(\text{'water'}),$$

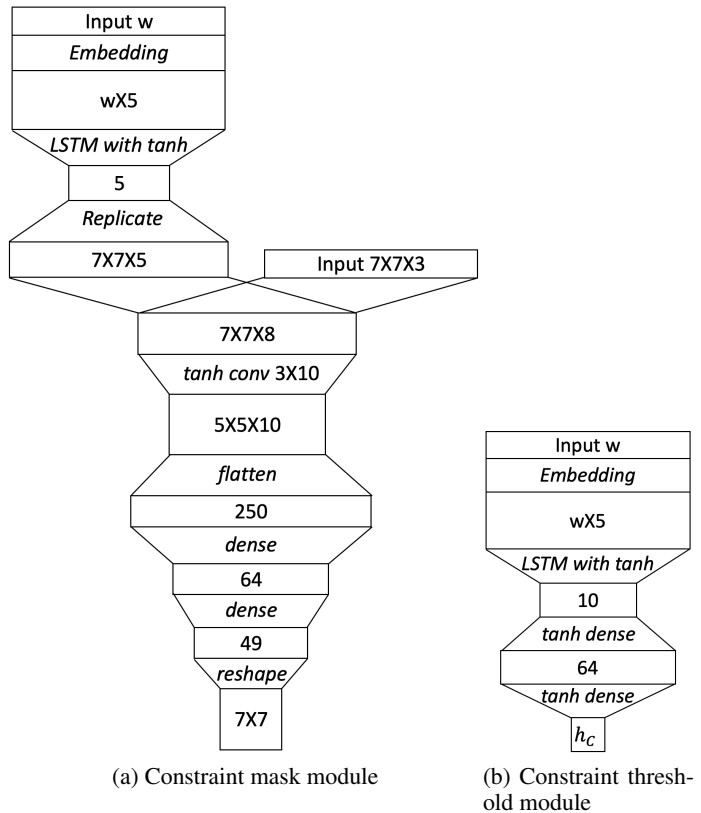

(a) Constraint mask module

(b) Constraint threshold module

Figure 8: Description of the constraint interpreter.

$$P(\text{`}lava'|\text{text constraints}) \propto P(\text{text constraints})P(\text{`}lava')$$
$$= \Pi_{\text{word}}P(\text{word}|\text{`}lava')P(\text{`}lava'),$$

$$P(\text{`}grass'|\text{text constraints}) \propto P(\text{text constraints})P(\text{`}grass')$$
$$= \Pi_{\text{word}}P(\text{word}|\text{`}grass')P(\text{`}grass').$$

Finally, we select the one with the maximum value to predict the cost entity.

**Details of the algorithm–PCPO.** We use a KL divergence projection in PCPO to project the policy onto the cost constraint set since it has a better performance than $L_2$ norm projection. We use GAE-$\lambda$ approach (Schulman et al., 2015b) to estimate $A_R^\pi(s,a)$ and $A_C^\pi(s,a)$. We use neural network baselines with the same architecture and activation functions as the policy networks. The hyper-parameters of training POLCO are in Table 4. We conduct the experiments on the machine with Intel Core i7-4770HQ CPU. The experiments are implemented in rllab (Duan et al., 2016), a tool for developing RL algorithms.

**Baseline model–Constraint Fusion (CF).** Our baseline is adapted from Walsman et al. (2018). The model is illustrated in Fig. 9. An LSTM takes the text $x$ as an input and produces a vector representation. The CNN takes the environment embedding of $o_t$ as an input and produces a vector representation. These two vector representations are concatenated, followed by a MLP to produce an action $a_t$. We do not consider other baselines in Janner et al. (2018) and Misra et al. (2018). This is because that their models are designed to learn a multi-modal representation (*e.g.,* processing a 3D vision) and follow goal instructions. In contrast, our work focuses on learning a constraint-satisfying policy.

The parameters of the baseline is shown in Fig. 10. We use the same CNN parameters as in our policy network to process $o_t$. Then, we use the same LSTM parameters as in our constraint mask

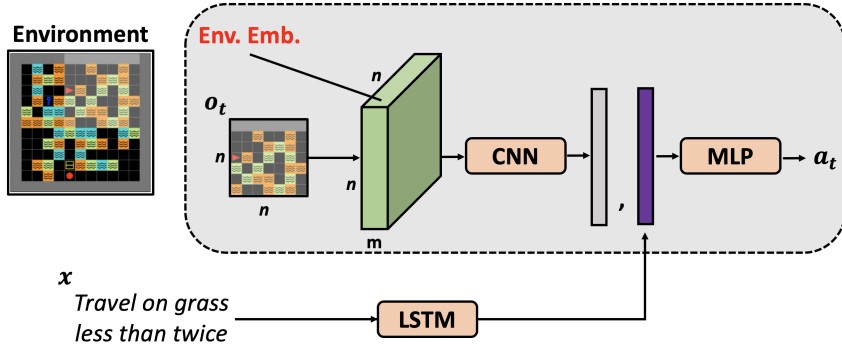

Figure 9: Baseline model–Constraint Fusion (CF). It is composed of two parts – **(1)** a CNN takes $o_t$ as an input and produce a vector representation, **(2)** an LSTM takes $x$ as an input and produce a vector representation. We then concatenate these two vectors, followed by a MLP to produce an action $a_t$.

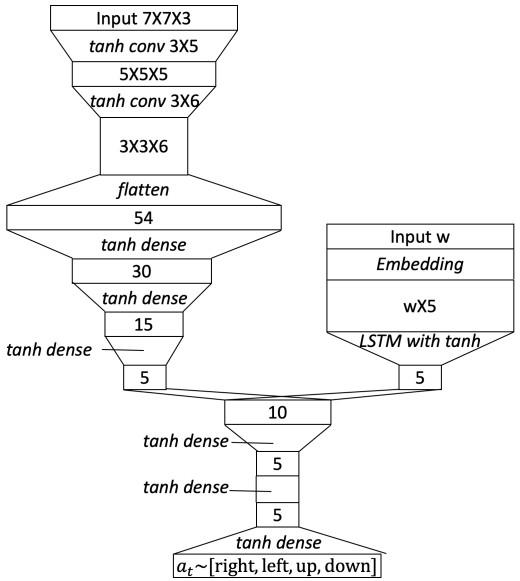

Figure 10: Description of our baseline model-Constraint Fusion (CF).

module to get a vector representation with size 5. Note that we use almost the same number of the parameters to ensure that POLCO does not have an advantage over CF. Finally, we use dense layers to the categorical distribution with four classes. We then sample an action from this distribution.

## A.3 ADDITIONAL EXPERIMENTS

**Learning curves of training the policy network.** The learning curves of the undiscounted constraint cost, the discounted reward, and the number of steps over policy updates are shown for all tested algorithms and the constrains in Fig. 11. Overall, we observe that

(1) POLCO improves the reward performance while satisfying the cost constraints during training in all cases,

(2) the policy network trained with TRPO has substantial cost constraint violations during training,

(3) the policy network trained with FPO is overly restricted, hindering the reward improvement.

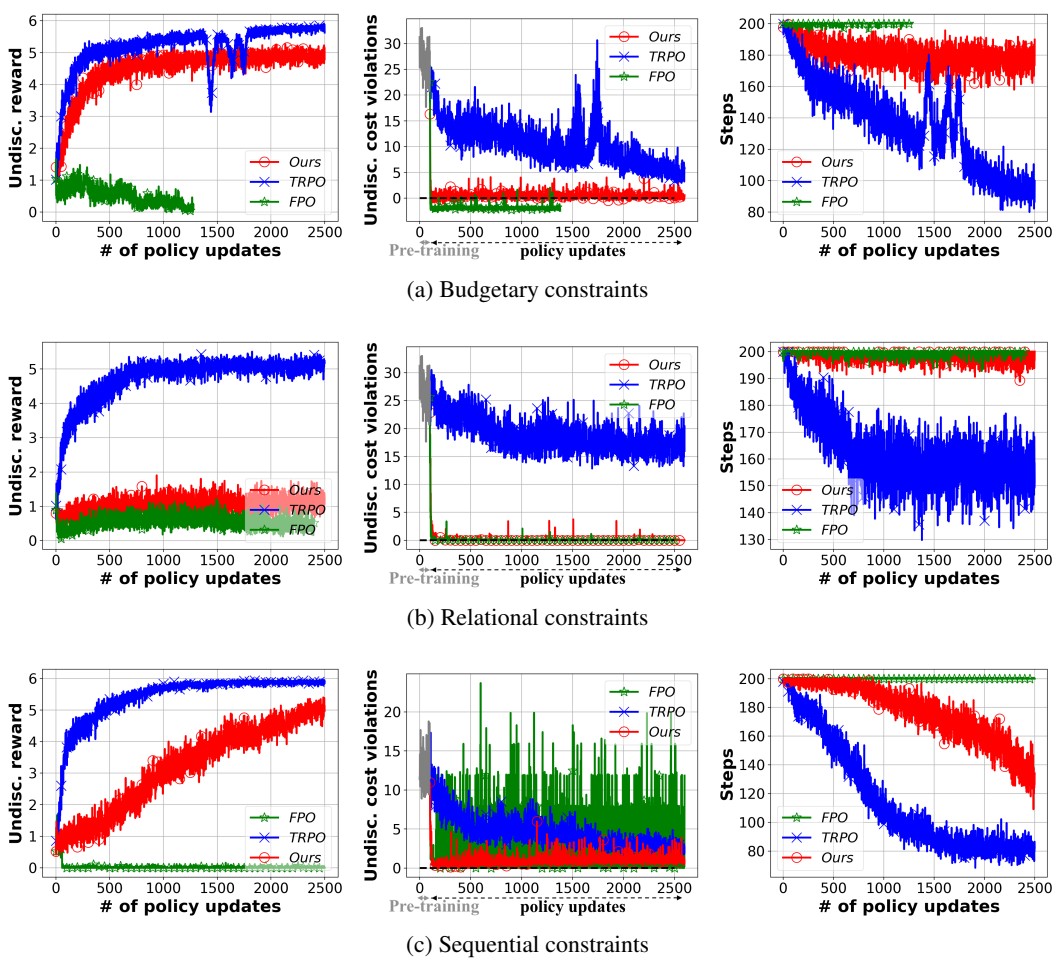

(a) Budgetary constraints

(b) Relational constraints

(c) Sequential constraints

Figure 11: **Learning curves of training the policy network.** The undiscounted reward, the undiscounted cost violations (*i.e.,* $\Delta_C = J_C(\pi) - h_C$), and the number of steps over policy updates for the tested algorithms and the constrains. In the undiscounted cost violations plots, we further include the numbers for the interpreter pre-training stage in the first 100 points. This is equal to 5000 trajectories. The maximum allowable step for each trajectory is 200. We observe that POLCO satisfies the cost constraints throughout training while improving the reward. In contrast, the policy network trained with TRPO suffers from violating the constraints and the one trained with FPO cannot effectively improve the reward. (Best viewed in color.)

## A.4   POLCO FOR PIXEL OBSERVATIONS/3D EGO-CENTRIC OBSERVATIONS

To deal with pixel observations $o_t$, we can still use the proposed architecture to process $o_t$ as shown in Fig. 12. To predict the cost constraint mask $\hat{M}_C$, we use the object segmentation method to get the bounding box of each object in the scene. As a result, the area of that bounding box will be one if there is a cost entity (*i.e.,* the forbidden states mentioned in the text). Otherwise, the bounding box contains a zero. For $\hat{M}_B$, we can use a similar approach to compute the cumulative cost violations at each step. In addition, to deal with navigation environments with 3D ego-centric observations, we propose shifting the $o_t$, $\hat{M}_C$ and $\hat{M}_B$ matrices to be the first-person view. The bounding box approach for image case can still be applied here. We leave this proposal to future work.

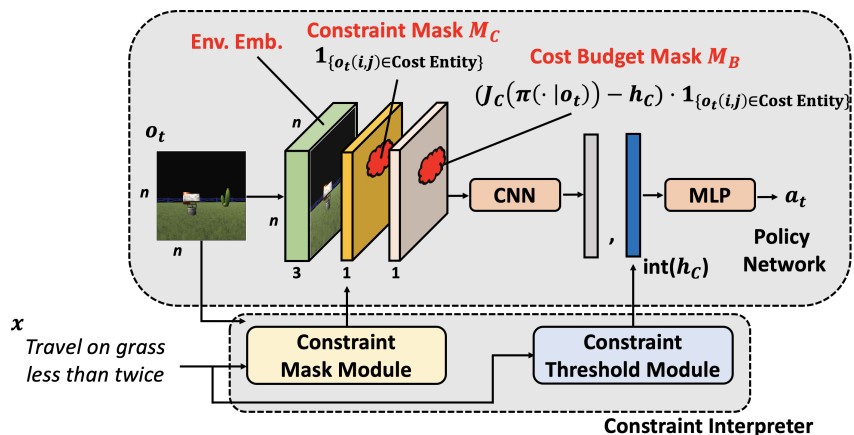

Figure 12: POLCO for pixel observations and 3D ego-centric observations. The red cloud area represents the bounding box of each object in $o_t$.

### A.5 THE MACHINE LEARNING REPRODUCIBILITY CHECKLIST (VERSION 1.2, MAR.27 2019)

For all models and algorithms presented, indicate if you include[2]:

- A clear description of the mathematical setting, algorithm, and/or model:
  - **Yes**, please see the problem formulation in Section 3, the algorithm and the model in Section 4.
- An analysis of the complexity (time, space, sample size) of any algorithm:
  - **Yes**, we provide discussions in Section 4 and Appendix A.2 for showing a training infrastructure.
- A link to a downloadable source code, with specification of all dependencies, including external libraries:
  - **Yes**, please see our abstract.

For any theoretical claim, check if you include:

- A statement of the result:
  - **Not applicable**.
- A clear explanation of any assumptions:
  - **Not applicable**.
- A complete proof of the claim:
  - **Not applicable**.

For all figures and tables that present empirical results, indicate if you include:

- A complete description of the data collection process, including sample size:
  - **Yes**, please see Appendix A.1 and Table 1 for sample size.
- A link to a downloadable version of the dataset or simulation environment:
  - **Yes**, please see our abstract.
- An explanation of any data that were excluded, description of any pre-processing step:
  - **Yes**, please see Appendix A.1.

---

[2]Here is a link to the list: `https://www.cs.mcgill.ca/~jpineau/ReproducibilityChecklist.pdf`.

- An explanation of how samples were allocated for training / validation / testing:
  - **Yes**, please see section 5 and Appendix A.1
- The range of hyper-parameters considered, method to select the best hyper-parameter configuration, and specification of all hyper-parameters used to generate results:
  - **Yes**, please see Appendix A.2.
- The exact number of evaluation runs:
  - **Yes**, please see section 5.
- A description of how experiments were run:
  - **Yes**, please see section 5.
- A clear definition of the specific measure or statistics used to report results:
  - **Yes**, please see Section 5.
- Clearly defined error bars:
  - **Not applicable.**
- A description of results with central tendency (*e.g.,* mean) variation (*e.g.,* stddev):
  - **Not applicable.**
- A description of the computing infrastructure used:
  - **Yes**, please see Appendix A.2.

