# OpenReview forum: "Safe Reinforcement Learning with Natural Language Constraints"
_ICLR.cc/2021/Conference — Reject_

### Official Review · AnonReviewer4 · 2020-10-17
**The research problem is interesting, but the scientific contribution of this paper is limited.**

**Rating:** 5
**Confidence:** 4

**Review:**

The paper proposed an algorithm to learn a policy when provided with natural language constraints. The paper defined a navigation task called Hazard World, in which an agent navigation on the map to collect items. The authors defined three types of constraints to restrict agents to visit certain states: 1. budgetary constraints, 2. relational constraints and 3. sequential constraints. The three constraints are described in natural language. The authors proposed a two-step solution. In step one, the algorithm learns a mapping between a natural language constraint to an intermediate representation. In step two the algorithm takes the intermediate representation to learn a policy that satisfy the constraints.

Pros:
1. The research problem here is interesting. Using natural language to specify constraints makes a better UI between humans and robots.
2. The way the authors categorize the constraints and the collected dataset by the authors are contributions to the research community.

Cons:
The scientific contribution of the proposed algorithm is rather limited. I don't see the uniqueness of the role played by natural language in the algorithm.

Detailed Comments:
The paper proposed to first learn a mapping from a natural language constraint to an intermediate representation (step 1), and then learn a policy that conditioned on this intermediate representation (step2). Natural language does not play any role in step 2; and step 1 is just a classification of natural language into a pre-defined structured representation (the mask and the threshold). The constraints do not need to be natural language, it can be images, video or anything. I understand that this could be a good thing, but what is the scientific contributions here? The idea of mapping natural language to structured representation is not new (e.g. to build chatbot, people map language to structured dialogue states, and then learn policies conditioned on dialogue states), and the idea of using projection-based constraints policy optimization is not new (since the authors use a existing work from Yang et al., 2020b).
I think the paper would be stronger (in terms of scientific contribution) if it makes some efforts on the following direction:
1. How would we learn a policy without the labels for the constraints interpreter. (The author mentioned that their model is end-to-end but due to sample efficiency they pre-train the interpreter with labels)
2. If labels for the constraints interpreter is a must, then how well does it generalize to new environment and new natural language constraints, and how robust is the optimization in step 2 with respect to the errors in step 1. Figure 5 partially answers this question. However, it contains only one new environment, and the natural language constraints were selected from Hazard World, which was used to train the interpreter.
3. Make use of the characteristics of the natural language when designing the algorithm. Currently, the natural language does not play an important role in the algorithm. The language constraints are simply encoded by a LSTM and feed to a set of classifier. However, natural language is compositional. It is possible that there are more than one constraints and thresholds in one sentence. How to design a sample efficient algorithm for this scenario? Currently the algorithm only supports one constraint/threshold at one episode.
4. How would we learn a policy if the cost C(s_t, a_t) is unobservable at each step (so that the agent has to infer the cost from natural language constraints).

These are just some thoughts that I feel that may be interesting, but of course, the authors would have a better idea of what is a better follow up direction for this work. My main point is that I feel that the authors haven't exploited enough the uniqueness of this "safe reinforcement learning with natural language constraints" problem, and the proposed solution is rather straight forward. However, I do acknowledge that this is a very hard problem and the authors made a nice step tackling this problem.

Question:
In Section 4.2 the authors mentioned that "Our policy model described in the previous section is end-to-end differentiable". How do you backpropagate through the threshold h_C if it is trained end-to-end?

---

> ### Author Response · Authors · 2020-11-13
> **Response to R4 - Part 1**
>
> We thank Reviewer #4 for the helpful and insightful feedback. We provide answers to individual questions below.
>
> Q1. Step 1 is just a classification of natural language into a pre-defined structured representation (the mask and the threshold). The constraints do not need to be natural language, it can be images, video or anything. I understand that this could be a good thing, but what are the scientific contributions here? The idea of mapping natural language to structured representation is not new (e.g. to build chatbot, people map language to structured dialogue states, and then learn policies conditioned on dialogue states), and the idea of using projection-based constraints policy optimization is not new (since the authors use a existing work from Yang et al., 2020b).
>
> Ans: Please see general response for a summary of our novel contributions.
> To reiterate, our scientific contributions are in defining a new task, developing a benchmark along with crowdsourced human-generated text, and providing a first foray into the modeling approach. We believe that this will create a new direction for future research and have application in several downstream real-world use cases where humans wish to control agents via language.
>
> Q2. How would we learn a policy without the labels for the constraints interpreter. (The author mentioned that their model is end-to-end but due to sample efficiency they pre-train the interpreter with labels)
>
> Ans: This is a challenging problem. Without the pre-training process, it is non-trivial to compute and backpropagate the gradient through $M_B$, which needs to compute the cumulative cost violations. In addition, we have tried an end-to-end approach called constraint-fusion (CF) shown in the experiment section. We found that the performance is bad since the agent does not have a good representation of the cost in the beginning. One way is to use something like a curriculum learning approach, in which we iteratively update the policy and the cost constraint interpreter using the reward and the cost signals. We leave these explorations to future work.
>
> Q3. If labels for the constraints interpreter is a must, then how well does it generalize to new environment and new natural language constraints, and how robust is the optimization in step 2 with respect to the errors in step 1. Figure 5 partially answers this question. However, it contains only one new environment, and the natural language constraints were selected from Hazard World, which was used to train the interpreter.
>
> Ans: We clarify that we have split our dataset into training and test sets (with unseen map environments) as mentioned in Section 5. Please note that the language constraints used in the test set are not seen in training. All the numbers reported in the paper are based on the test set, which shows the generalization ability of the model.
>
> Q4. Make use of the characteristics of the natural language when designing the algorithm. Currently, the natural language does not play an important role in the algorithm. The language constraints are simply encoded by a LSTM and feed to a set of classifier. However, natural language is compositional. It is possible that there are more than one constraints and thresholds in one sentence. How to design a sample efficient algorithm for this scenario? Currently the algorithm only supports one constraint/threshold at one episode.
>
> Ans: We emphasize the goal of the paper is to formulate a new problem and create a new dataset to understand how we can deploy instruction following systems in real applications with the consideration of the safety. We agree that right now we only consider one type of the constraints. We hope more researchers can work on this problem and leave this for future work to improve.
>
> Q5. How would we learn a policy if the cost C(s_t, a_t) is unobservable at each step (so that the agent has to infer the cost from natural language constraints).
>
> Ans: The ground truth cost $C(s_t, a_t)$ is unobservable when training the policy. The agent navigates using inferences from the constraint interpreter. We believe we caused confusion by abusing the notation for C. In Figure 2, $C(s_t, a_t)$ is an inference from the constraint interpreter. We will make this clear in our revision.
>
> Q6. My main point is that I feel that the authors haven't exploited enough the uniqueness of this "safe reinforcement learning with natural language constraints" problem, and the proposed solution is rather straight forward.
>
> Ans: Please see the response to Q4.

---

> > ### Author Response · Authors · 2020-11-13
> > **Part 2**
> >
> > Q7. In Section 4.2 the authors mentioned that "Our policy model described in the previous section is end-to-end differentiable". How do you backpropagate through the threshold h_C if it is trained end-to-end?
> >
> > Ans: Our model architecture does allow for end-to-end differentiability via tricks like Gumbel Softmax (one can treat the prediction of $h_C$ as a continuous variable). However, in this version, we do not train end-to-end for computational reasons. We will add a discussion on this to the paper.

---

### Official Review · AnonReviewer2 · 2020-10-23
**Hybridization of two problems (NLU and Safe RL) is not clearly a new problem**

**Rating:** 6
**Confidence:** 3

**Review:**

This paper present an experiment of safe reinforcement on a 2D grid-word where the safety constraints are specified in natural language instead of being specified formally. The justification of this system is to allow non-experts to train agents using safe-RL.
According to the authors: "The key challenge lies in training the agent to interpret natural language and naturally adhere to the constraints during exploration and execution".
The proposed system is made of two parts: a constraint interpreter that is (mostly) trained in a supervised way with Amazon Mechanical Turk to translate natural language orders into grid-world ad-hoc constraints and a policy that is trained through PCPO, a TRPO-like constraint-aware policy optimization algorithm.
That's certainly a nice piece of engineering, but honestly my first sentiment when reading the motivation of this paper was astonishment. If the agent is already able to understand complex natural language statements about its environment during its exploration/training phase, why would it need further training to apply these constraints on a simple grid-world ?

If the natural language understanding (NLU) task has to be handled before the agent's exploration/training phase, we are facing a concatenation of two problems: NLU then Safe-RL, not a new problem involving tightly NLU and safe-RL.
Even if some publications were already made on that topic, training a agent to properly follow natural language orders, that may include constraints, during its execution phase is yet an unsolved problem that requires a real fusion between NLU and RL that is of scientific interest.

---

> ### Author Response · Authors · 2020-11-13
> **Response to R2**
>
> We thank Reviewer #2 for the helpful and insightful feedback. We provide answers to individual questions below.
>
> Q1. If the agent is already able to understand complex natural language statements about its environment during its exploration/training phase, why would it need further training to apply these constraints on a simple grid-world?
>
> Ans: We believe there is some misunderstanding of our setup here and apologize for the inconvenience. Our model contains two components: (1) a constraint interpreter to process the text to understand the constraints; (2) a policy network that trained with RL to produce safe actions given by the text constraints.
>
> The natural language statements only describe safety constraints for the tasks. They do not say anything about how to collect rewards and accomplish the task itself. Even if an agent “understands” these statements apriori (via pre-training), we require further training of the policy (adhering to the constraints) to learn to complete the tasks. Reward signals and goal states are not described by natural language here (in contrast to traditional instruction following setups)
>
> *Importantly*, once all the model components are trained, there is no further change to the parameters. During execution, the agent is set in a random environment and provided a randomly sampled constraint to follow (e.g., Don’t touch grass more than two times). We simply measure the ability of the policy to complete the task without violating the constraints.
> Hope that clarifies the motivation of the problem. Please let us know if you still have questions on this aspect!
> We will also update the paper to clarify this better.
>
> Q2. If the natural language understanding (NLU) task has to be handled before the agent's exploration/training phase, we are facing a concatenation of two problems: NLU then Safe-RL, not a new problem involving tightly NLU and safe-RL.
>
> Ans: We respectfully disagree since our approach does in fact jointly consider the language interpretation and policy learning. Note that despite the pre-training, the constraint interpreter is not perfect and the policy network has to deal with the uncertainty in the constraint predictions while navigating the environment. Thus, the policy network’s training is not independent of the constraint interpreter.
>
> Moreover, the ‘concatenation’/pipelined approach is a feature of our model, not the task we introduce itself. One is totally free to design a model that utilizes the different signals (constraints and observations) in other ways.
>
> Further, while NLU+RL has been explored before, we know of no previous work in the space of NLU for specifying constraints in Safe-RL (our problem). Considering navigation constraints that are orthogonal to reward requires new approaches distinct from instruction following or other NLU+RL domains. To our knowledge, approaches for Safe-RL have not incorporated natural language at our scale. Further, we demonstrate that existing approaches to instruction following that process both text and state observations through simple operations like concatenated representations do not work well for handling constraints. Instead, translating the text to structured forms ($M_C, M_B, h_C$) makes it more useful and effective for safe RL.
>
> Please also see the general response for more discussion on our main contributions.

---

### Official Review · AnonReviewer3 · 2020-10-28
**Official Blind Review #3**

**Rating:** 5
**Confidence:** 3

**Review:**

This paper presents a new test environment, Hazard World, for learning the safe reinforcement learning agents with given natural language constraints. In this problem, the goal of the agent is to find an optimal policy that maximizes the cumulative rewards while satisfying the constraints given in natural language. The authors introduce the model that contains the following two separate components; constraint interpreter for encoding the language constraints and policy network for learning the RL agent. Finally, they report the results of their proposed algorithm and compare it with the baselines.

This paper is well organized overall, but I have several concerns and questions about the paper.
- In section 3 problem formulation, the authors formulate the problem as partially observable constrained MDP, and assume that the agent does not know the constraint specification $C$. However, in page 4, the description of policy network, it appears that $C$ is used when calculating the $M_B$ matrix. If $C(s_t,a_t;x)$ is assumed to be given from the environment, this is considered to be a strong assumption. I would like to receive detailed answers from the authors on this part.
- In Page 4, last paragraph of constraint interpreter (“For the sequential constraints with … ”), I don't understand what this paragraph exactly means. Does that part contain more detailed considerations, rather than just using the LSTM to handle the sequential constraints?
- (Probably related to the first question) I think that the budgetary constraints should consider history as well as sequential constraints. For example, considering the budgetary constraints in Figure 1, it is not possible to determine whether the constraint has been violated with only observation $o_t$, and it is possible only with the history information. Also, if this is because $C(s_t,a_t;x)$ is provided from the environment as in the first question, I think that it is not necessary to use LSTM for sequential constraints as well.

For the experiments,
- In Table 2 (a)-(c), authors compare the performance of methods with $\Delta_{RC}$. However, $\Delta_{RC}$ depends on the $\alpha$ value, and an appropriate alpha cannot be determined. Moreover, reward and cost represent completely different values and have different scales. I think it is inappropriate to use $\Delta_{RC}$ as an evaluation metric.
- It seems more natural to compare with Constrained Policy Optimization (CPO) than with FPO as a baseline algorithm. Is there any special reason for using FPO as a baseline instead of CPO? And, in the case of FPO, the result is likely to be very different depending on the value of fixed penalty, what values did you experiment with fixed penalty?
- In Figure 4 (b), I don't understand why the agent's view appears like that figure (Why is most of the area represented in black and the agent position does not appear?).
- In the POLCO algorithm, the constraint interpreter was pre-trained with collected data, but I wonder the result without the pre-training process. Also, for the results of baseline algorithms, are they also the results of using data equally? (including the pre-training process)

The main difference between the problem proposed in this paper and the existing constraints reinforcement learning problem is that constraints are given in the natural language. However, the author simply pre-trained a constraints interpreter that encodes a given constraint in natural language using additionally collected data without considering the RL point of view for this part, and combined it with the one of the existing constrained reinforcement learning algorithm PCPO. Therefore, although I acknowledge the considerable work for introducing a new safe RL environment with natural language constraints, I think that the contribution is insufficient unless consideration of natural language constraints is added.

---

> ### Author Response · Authors · 2020-11-13
> **Response to R3 - Part 1**
>
> We thank Reviewer #3 for the helpful and insightful feedback. We provide answers to individual questions below.
>
> Q1. In section 3 problem formulation, the authors formulate the problem as partially observable constrained MDP, and assume that the agent does not know the constraint specification. However, in page 4, the description of policy network, it appears that C(s_t,a_t;x) is used when calculating the M_B matrix. If C(s_t,a_t;x) is assumed to be given from the environment, this is considered to be a strong assumption. I would like to receive detailed answers from the authors on this part.
>
> Ans: $C(s_t,a_t;x)$ is a prediction from the constraint interpreter. This point was unclear in the paper; we will fix it. When computing $M_B$ for the policy network, we use the constraint interpreter to predict $C(s_t,a_t;x)$ and the threshold $h_C$ and keep a running total of the costs up to a given step. We do not use a ground-truth cost signal from the environment.
>
> Q2. In Page 4, last paragraph of constraint interpreter (“For the sequential constraints with … ”), I don't understand what this paragraph exactly means. Does that part contain more detailed considerations, rather than just using the LSTM to handle the sequential constraints?
>
> Ans: We use the LSTM to keep track of states visited by the agent to handle sequential constraints. The rest of the constraint interpreter is the same as the constraint interpreter for budgetary and relational constraints. There are no additional considerations.
>
> Q3. (Probably related to the first question) I think that the budgetary constraints should consider history as well as sequential constraints. For example, considering the budgetary constraints in Figure 1, it is not possible to determine whether the constraint has been violated with only observation $o_t$, and it is possible only with the history information. Also, if this is because $C(s_t,a_t;x)$ is provided from the environment as in the first question, I think that it is not necessary to use LSTM for sequential constraints as well.
>
> Ans: Yes, the budgetary constraint also considers the history. The history of the budgetary constraints (i.e., how much of a budget has been used) is embedded in $M_B$, via the $C(s_t,a_t;x)$ predicted by the constraint interpreter. We will make it clearer in the paper.
>
> On the other hand, the LSTM for the sequential constraint is used to keep track of the states visited by the agent to determine whether certain conditions have been satisfied or not. For example, the constraint “do not visit A after visiting B.” We should know whether the agent has visited B or not in order to determine whether A is a forbidden state. In addition, $C(s_t,a_t;x)$ is not provided from the environment. Hence, we need a network with memory.
>
> Q4. In Table 2 (a)-(c), authors compare the performance of methods with $\Delta_{RC}$. However, $\Delta_{RC}$  depends on the $\alpha$ value, and an appropriate $\alpha$ cannot be determined. Moreover, reward and cost represent completely different values and have different scales. I think it is inappropriate to use $\Delta_{RC}$ as an evaluation metric.
>
> Ans: We agree that the reward and the cost values are different units. However, this metric can reflect how we tradeoff between the reward and the cost. The value of $\alpha$ represents the trade-off preference for the task and depends on the use case -- if safety is of utmost importance, then $\alpha$ will be high. For example, if a constraint violation means breaking a robotic arm, then one would want a policy that does well when evaluated with a high $\alpha$.
>
> Q5. It seems more natural to compare with Constrained Policy Optimization (CPO) than with FPO as a baseline algorithm. Is there any special reason for using FPO as a baseline instead of CPO? And, in the case of FPO, the result is likely to be very different depending on the value of fixed penalty, what values did you experiment with fixed penalty?
>
> Ans: The comparison of FPO is to show the straightforward solution of solving this safe RL with text constraints problem by simply treating the cost as a negative reward. FPO requires hyperparameter tuning to find the best penalty value. We did a grid-search and found a penalty of 1 achieved the best performance. The base optimization problem of CPO and PCPO are similar and we provide results for PCPO (Our model is trained with PCPO shown in Table 2). Hence, we choose FPO to highlight the point that we cannot treat the cost as the negative reward.

---

> > ### Author Response · Authors · 2020-11-13
> > **Part 2**
> >
> > Q6. In Figure 4 (b), I don't understand why the agent's view appears like that figure (Why is most of the area represented in black and the agent position does not appear?).
> >
> > Ans: This is because the agent happens to face the wall in the first two steps. There is no object behind the wall, so the view is black. We reiterate that we consider a partially observed setting where the agent can only observe a 7x7 grid in the direction it is facing. We will make this clearer in the paper.
> >
> > Q7. In the POLCO algorithm, the constraint interpreter was pre-trained with collected data, but I wonder the result without the pre-training process. Also, for the results of baseline algorithms, are they also the results of using data equally? (including the pre-training process)
> >
> > Ans: Without the pre-training process, it is non-trivial to compute and backpropagate the gradient through $M_B$, which needs to compute the cumulative cost violations. We believe this is a challenging problem for future research on this benchmark task.
> > Yes, the data is used equally. For all the algorithms, including all baselines, we use the same training and test splits, including pre-training where appropriate. Pre-training data is bootstrapped via exploration on the training data.
> >
> > Q8. The main difference between the problem proposed in this paper and the existing constraints reinforcement learning problem is that constraints are given in the natural language. However, the author simply pre-trained a constraints interpreter that encodes a given constraint in natural language using additionally collected data without considering the RL point of view for this part, and combined it with the one of the existing constrained reinforcement learning algorithm PCPO. Therefore, although I acknowledge the considerable work for introducing a new safe RL environment with natural language constraints, I think that the contribution is insufficient unless consideration of natural language constraints is added.
> >
> > Ans. We respectfully disagree since our approach does jointly consider language interpretation and policy learning. Note that despite the pre-training, the constraint interpreter is not perfect and the policy network has to deal with that uncertainty while navigating the environment. Thus, the policy network’s training is not independent of the constraint interpreter.
> >
> > The reason we do not use a fully end-to-end training approach or learning text constraints and using RL at the same time is that we cannot naively backpropagate the gradient through $M_B$, which needs to compute the cumulative cost violations. In addition, without $M_B,$ we have shown that the performance is worse in the ablation study (Table 2(d)). Finally, we show that the proposed model outperforms other baselines which are commonly used in the literature of instruction following. Please also see the general response for more discussion on our contributions.

---

> > > ### Comment · AnonReviewer3 · 2020-11-25
> > > **Response to rebuttal**
> > >
> > > Thank you for taking the time to clarifications and considering my comments. Most of my questions and concerns were resolved by rebuttal and revision. The paper is well presented and I acknowledge again the considerable work for introducing a new safe RL environment with natural language constraints. However, It still seems that the contribution of this paper is limited except for suggesting a new environment. So I will keep my score.

---

### Official Review · AnonReviewer1 · 2020-10-30
**Important problem; Novel formulation**

**Rating:** 7
**Confidence:** 4

**Review:**

Summary:
The paper addresses how to learn policies for tasks in which constraints are specified in natural language. Towards this, the paper proposes a model that encodes the different types of natural language constraints into intermediate representations that model both spatial and temporal information between states. Then, they use this as input along with the observation to produce an action at each time step for a safe trajectory.
They also propose a new benchmark (Hazard World) which is inspired by the 2D MiniGrid environment. They show the efficacy of their models on this new benchmark outperforming several baselines.

Strengths:
1. The authors address an important problem of modeling natural language constraints for safe RL. Unlike previous work, which requires manual specification of constraint or rule bases constraint, specifying natural language constraints is more intuitive and scalable.
2. The proposed model which breaks down the language constraint into a constraint mask and constraint threshold is intuitive. The experiments also demonstrate
3. The authors also propose an interesting benchmark to evaluate methods for safe RL under natural language constraints. Even though the environment is built on top of 2D grid world with simple action spaces, constraints are still specified in free-form natural language. The benchmark will be useful for the wider RL community.
4. The paper is well written and easy-to-read.


Weaknesses:
1. The proposed models seem to be engineered for the 2D grid-like environments used for evaluation. I'd be interested in knowing how to generalize the models to other types of environments and observations (for instance navigation environments with 3D ego-centric observations). The authors mention that they describe an extension to 3D scene inputs in the appendix but I didn't find those details. Please let me know if I missed it in the appendix.

2. Mapping constraints into the current observation is a limiting assumption. Can all constraints be mapped to current observations?
 - 2.1 How can the proposed model deal with constraints that might be violated due to partial observability? For instance, if the lava is behind the agent's field of view, how will the agent incorporate those constraints.
 - 2.2 There might be other temporal constraints that need to be modeled. For instance, defuse a bomb in less than 5 steps like constraints that don't necessarily describe a visual constraint. How can a model handle such constraints?

---

> ### Author Response · Authors · 2020-11-13
> **Response to R1**
>
> We thank Reviewer #1 for the helpful and insightful feedback. We provide answers to individual questions below.
>
> Q1. The proposed models seem to be engineered for the 2D grid-like environments used for evaluation. I'd be interested in knowing how to generalize the models to other types of environments and observations (for instance navigation environments with 3D ego-centric observations). The authors mention that they describe an extension to 3D scene inputs in the appendix but I didn't find those details. Please let me know if I missed it in the appendix.
>
> Ans: You are correct in that the current experiments are in 2D environments. However, we believe that the framework can be easily extended to 3D and other cases, and will require appropriate architectures to process the different input forms. Sorry, we seem to have missed adding this discussion in the submitted version and will update the paper accordingly.
>
> For the 3D scene, as an example, we could use another convolution neural network to downsample the inputs and learn a representation for the environment. The constraint mask $M_C$ in this case would be an attention map that predicts how likely there is a cost entity in the scene.
>
> Q2. Mapping constraints into the current observation is a limiting assumption. Can all  constraints be mapped to current observations?
>
> Ans: The constraints considered in the paper are based on observational constraints, i.e., avoiding certain states. Based on this, our model is feasible for any environment. For example, in 3D, one could use SLAM to create the equivalent of our 2D world view and localize constraints into 3D objects. Further, note that we do not always consider just the current observation. For example, in sequential constraints, the agent has to reason over the past states as well.
>
> That said, you are right in that there could be constraints that are dependent on other aspects of the MDP (e.g., actions or abstract strategies). Exploring such constraints in the future would be an interesting extension of our work.
>
> Q3. How can the proposed model deal with constraints that might be violated due to partial observability? For instance, if the lava is behind the agent's field of view, how will the agent incorporate those constraints.
>
> Ans: In our experimental setting, the agent will never violate a constraint unknowingly. To discuss your example, in order to visit lava that is behind the agent’s field of view, the agent must turn around first (there is no “step back” action). In other words, the agent will observe the lava before taking an action.
>
> For other environments, this does raise an interesting question. Our model does not produce an explicit map of the environment or explicitly store past states in memory, which might be necessary to solve environments where the agent can violate constraints due to partial observability. We leave this for future work.
>
> Q4. There might be other temporal constraints that need to be modeled. For instance, defuse a bomb in less than 5 steps like constraints that don't necessarily describe a visual constraint. How can a model handle such constraints?
>
> Ans: In our setting, we consider constraints as information for the agent on what not to do. “Defuse a bomb in less than 5 steps” incorporates an instruction, in that the utterance also defines the agent’s goal. Combining instructions with constraints is an interesting future direction!
>
> We do not have non-visual constraints (e.g., constraints that are purely temporal or purely based on action sequences) in our dataset. To handle such constraints, we could extend our model by designing a NN that outputs the representation for representation constraints, and use that for the policy network.

---

### Author Response · Authors · 2020-11-13
**General response/clarification for the contribution and the proposed approach**

We thank all the reviewers for their thorough and helpful feedback.

**Contributions**

We would like to restate our main contributions, which to our knowledge, make our work novel:
*We propose the problem of specifying constraints for safe reinforcement learning in natural language.* This has not been attempted in prior work to the best of our knowledge. As mentioned by all reviewers, this problem is important since specifying natural language constraints is more intuitive and scalable, and is beneficial for designing personalized robotic or instruction following systems.

Note that this is *quite different* from traditional instruction following since the text tells the agent what *not to do* - beyond following the rules/constraints, the agent is free to learn any policy to actually accomplish its task, which is not specified in the text. We create a new benchmark task (Hazard World) containing three broad classes of abstract safety criteria and diverse free-form text crowdsourced from Amazon Mechanical Turk (no templated or synthetic language).

Finally, we propose one solution (by no means the best one!) to solve the problem and empirically compare our model against other baselines which are commonly used in the literature of instruction following. Even though our approach requires pre-training the constraint interpreter with a few samples, we demonstrate that integrating language understanding into modern safe RL algorithms is possible and better than simply concatenating the text representation and the observation with end-to-end training.

**Learning process**

We wish to clarify any potential misunderstandings to the learning setup (R2, R4).
There are two learning problems in our setup: (1) mapping natural language to constraints, and (2) learning a policy with constraints (which in itself is a challenging problem). We pre-train the constraint interpreter (1) to make training easier. Even with (1) a pre-trained interpreter (which is imperfect), the agent must learn a policy to solve tasks with varying constraints that are provided during each episode.

That said, our model is a single neural architecture that can take both state observations and the constraint specification in text as inputs and output a policy. Training this model end-to-end is possible using tricks like Gumbel softmax to handle backpropagation through discrete intermediate outputs like $M_B$. We leave this to future work.

We will upload an updated version. In the meantime, we have provided detailed responses to all your questions below - please take a look and let us know if you have any further questions/comments.

---

> ### Author Response · Authors · 2020-11-17
> **Uploaded the Revised Version**
>
> We have uploaded an updated version (the modified part is shown in red). Please take a look and let us know if you have any further questions/comments.
>
> We describe some of the changes in the paper here.
> (1) We use $\hat{M}_C$ and $\hat{M}_B$ to donate the *predicted* cost constraint mask and cost budge mask. The pre-training and the policy learning stages are entirely trained on the training set $\mathcal{D}_\mathrm{train}.$
> (2) We add the 3D ego-centric view discussions in the Appendix.

---

### Decision · Program_Chairs · 2021-01-07
**Final Decision**

**Decision:**

Reject

**Comment:**

The goal of the paper is to learn policies that can solve a given task while adhering to certain constraints specified via natural language. The paper closely builds upon prior work on constrained RL and passes the representation of natural language constraints by pre-training an interpreter. Experiments are done in a new proposed 2D grid-world benchmark. Although reviewers liked the premise, the main issue raised is that the way natural language constraints are handled is no different from the way it is done in prior work on constrained RL. The authors provided the rebuttal and addressed some of the concerns regarding paper details. However, upon discussion post rebuttal, the reviewers and AC feel that the paper does not provide clear scientific insight because the natural language part is processed separately from the policy learning part. We also believe that the paper will immensely benefit with results in more complex environments beyond the 2D grid-world. Please refer to the reviews for final feedback and suggestions to strengthen the submission.